# Impact of Microclimate on People's Experiences and Behaviours in the Cultural Consumption Space: A Case Study of Panjiayuan Antique Market in Beijing, China

**Mo Han** [1,2,3], **Bing Han** [1,3], **Siyi Liu** [4] **and Ziwen Sun** [1,2,3,*]

1   School of Design and Arts, Beijing Institute of Technology, Beijing 102401, China
2   Tangshan Research Institute, Beijing Institute of Technology, Tangshan 063099, China
3   Social Space and Healthy Environment Lab, Beijing Institute of Technology, Beijing 102401, China
4   Academy of Arts and Design, Tsinghua University, Beijing 100190, China
*   Correspondence: ziwen.sun@bit.edu.cn

**Abstract:** Antique and cultural consumption spaces make a great contribution to urban vitality where numerous people walk, stay, and trade. However, how these people's experiences and behaviours are affected by the microclimate of such spaces has not been studied till now. To address this gap, and using the concept of Post-occupancy Evaluation (POE) as a basis, our study investigated microclimate factors, subjective experiences, and spatiotemporal behaviour patterns in Panjiayuan Antique Market in Beijing, China. Using a mixed-methods approach comprising subjective questionnaires (n = 101), face-to-face interviews (n = 81), spatiotemporal behaviour mapping (n = 8455), and on-site observations, our results showed that microclimates impact people's experiences and behaviours, with visibility and noise being the two primary impact factors. Most female visitors are more sensitive to the microclimate than male visitors. Furthermore, vendors in the Antique Market preferred to amend their nearby environments to increase the microclimate quality to offer a better experience to visitors around them. This study developed a comprehensive methodology that expands POE in relation to microclimatic factors in the context of cultural consumption spaces. These findings suggest that microclimates have different impacts on people's experiences and behaviours in different spaces, which should be considered when designing and renewing urban antique markets in the future.

**Keywords:** microclimate; urban experience; walking behaviour; spatiotemporal patterns; post-occupancy evaluation (POE)





## 1. Introduction

Post-occupancy Evaluation (POE) refers to a series of rigorous evaluations of buildings that have been built and used for a certain duration, to test whether the buildings still meet the needs of the occupants and their performance. The goal of POE is to identify problems or potential factors that could lead to the building not serving its purpose, and propose solutions for different levels of development, so that the building can be used properly for a longer duration while providing a better experience for people [1]. Indoor environmental comfort is an important dimension in the POE of buildings because it can directly influence people's experiences and indirectly influence their behaviour [2]. Since the COVID-19 pandemic, outdoor public spaces have become increasingly popular for urban residents in China. For example, vending markets are one of the most vital space types in cities [3]. Microclimate is a key factor influencing the use of urban outdoor spaces, and people often choose a comfortable microclimate for outdoor activities. A good urban outdoor space can promote spontaneous social activities, thereby increasing the efficiency of urban space use and enhancing urban vitality. However, most POE studies have ignored outdoor microclimates, which may have different impacts on human experience and behaviour.

Microclimates are an important part of urban ecology, which can be improved by appropriate planning and design. The term climate can be classified as macroclimate, mesoclimate, regional climate, and microclimate, according to scale [4]. A microclimate is defined as part of the ground boundary layer, where temperature and humidity are influenced by ground vegetation, soil, and topography [5]. The concept of microclimate as "a small-scale form of climate formed by climatic deviations in a specific area within a few kilometers of different plots" [6]. According to Fu, microclimates are completely different from the general climate of the region owing to the influence of local factors in individual places in the near-surface atmosphere [7]. Many scholars have recognised that small-scale characteristics of microclimates have more microscopic indicator parameters, and that a microclimates are a climatic condition in a small area influenced by urban climate and regional morphology, and are a vital factor in environmental equality [8].

Microclimate characteristics can be reflected by physical parameters; presently, the physical parameters of interest primarily include air temperature, solar radiation, relative humidity, and wind speed [9]. These parameters are the major indicators for measuring microclimate in most cases. Based on these four indicators, microclimate data can be obtained using two primary methods; namely, subjective questionnaires, and software simulations combined with field measurements. Software simulations combined with field measurements focus on simulating real environmental conditions and environmental performance analysis [10]. Subjective questionnaires focus on the combined analysis of individual factors and microclimate effects, and on the correlation between microclimates and outdoor thermal comfort [11].

Most studies on microclimates have focused on the relationships between microclimates, urban morphology, and landscape greenery, such as the regulation of microclimates by street morphology [12,13], urban morphology [14,15], spatial layout [16], and vegetation [17]. In addition, a systematic literature analysis was conducted in the fields of microclimate and thermal comfort, and the development of this field is well established [18]. The outdoor thermal environment directly affects outdoor activities and the use of outdoor space [11]. Hu, Li, and Chen categorised assessment of the thermal environment into subjective and objective factors. Subjective factors include social habits and behavioural patterns, and are investigated using questionnaires, observations, and interviews. Objective factors include climatic conditions and physiological regulation and are investigated through measurements and simulations [18]. However, people often have more varied thermal experiences and expectations outdoors than in stable indoor thermal environment [19]. Recent studies indicate that environmental psychology is an alternative way of understanding microclimate impacts [20]; namely, that microclimate factors (e.g., spatial features, climate, and weather) might carry different meanings and have various impacts on human behaviours in different situations and with cultural backgrounds, none of which has yet been sufficiently explored. In addition, studies on the relationship between microclimates and human behaviour have shown that there are four major microclimatic factors that influence behaviour in outdoor spaces; namely, air humidity, temperature, wind, and solar radiation [21–24]. The impact of urban spatial elements on the microclimate can be summarised at the macro- and microspatial levels. Studies at the microspatial level have mostly focused on the impact of microclimates on pedestrian flow and the intensity of human activity [25–32]. Studies on the impact of microclimates on behaviour have focused on three types of behaviour, including recreational activities, physical activity, and walking. Liang et al. suggested that activity facilities act as spatial attractions and trigger sudden changes in pedestrian walking speeds within a space at the level of individual intentions [31]. However, fewer studies have taken the time into account; namely, the impact of changes in microclimate on people's behaviour at different times of a day, and the differences in people's subjective perceptions of the same microclimate on weekdays versus weekends.

## 2. Research Questions and Aim

Among different market types, the antique and cultural market has strong sociodemographic characteristics, with various culture-related behaviours that differ from those of other outdoor spaces. However, these have been ignored in studies so far, and the impact of the microclimate of specific spatial features on behaviour has not yet been explored. Therefore, the aim of this study was to examine the relationships between individual factors, and the impact of the microclimate on spatial experience and human behaviour in Panjiayuan Antique Market, Beijing. Based on the above conception, the following research questions were framed:

- How does microclimate impact spatial experience and human behaviour in Panjiayuan Antique Market?
- Are the impacts of microclimate different between male and female visitors?

## 3. Methods

### 3.1. Study Area

The Panjiayuan Antique Market is a symbol of China's national and historical culture, having hosted dozens of foreign dignitaries, and is an important place for foreign and domestic tourists visiting Beijing. It is managed by state-owned enterprises in a unified manner and is the most extensive antiques market in the country, with several established cultural research institutions there. Since 1992, when a spontaneous antique market was gradually forming, the government officially positioned the market as a "cultural and creative industrial park" in the inner city of Beijing, proposing a plan to "retreat from the street into the market" in order to build the current Beijing Panjiayuan Antique Market. Major types of goods sold in the market include bronzes, the Four Treasures of the Study, Chinese jewellery and jade, porcelain, traditional handicrafts of ethnic minorities, Chinese furniture, stone forest carvings, antique paintings and calligraphy, and second-hand books. The Panjiayuan Antique Market is located near the East Third Ring Road in Beijing's Chaoyang District, just 10 km from Tiananmen Square, the political centre of the city. It is conveniently located near the Line 10 subway station, itself named after "Panjiayuan", in addition to about a dozen bus lines with special stops there. It covers an open area of about 48,500 m$^2$. According to statistics, Panjiayuan Antique Market has a daily passenger flow of up to 60,000 people. There are more than 4000 stalls, and close to 1000 vendors have set up their stalls here permanently.

Through field observations, four sites in the Panjiayuan Antique Market were selected for the study (Table 1): site 1, a western stall; site 2, an old book stall; site 3, ancient style shops; and site 4, an antique jumble area (a modern collection zone). Site 1 retained the original form of the ground stall, with an open space and only a soft canopy for shade and protection from the rain, and was located near one of the entrances to the Panjiayuan Antique Market, mainly selling old antique items and traditional handicrafts of ethnic minorities. Site 2 was located underneath an elevated bicycle park lot, with poor environmental quality—the ceiling was low and the space itself was dark—and mainly sold old books. Site 3 was a row of two-story high ancient style shops on one side and a five-story steel structure building on the back. It was like an alley without a roof, located near the main entrance of the Panjiayuan Antique Market, and mainly sold antique goods, such as old coins and watches. Site 4 was a semi-open shed space with stalls arranged regularly, and mainly sold strings, beads, literary treasures, calligraphy, and paintings.

**Table 1.** Characteristics of the four selected study sites in the Panjiayuan Antique Market.

| | Site 1<br>Western Stalls | Site 2<br>Old Book Stalls | Site 3<br>Ancient Style Shops | Site 4<br>Antique Jumble Area |
|---|---|---|---|---|
| Location | near the west entrance | northwest side of<br>the market | opposite the north entrance | the east sheds area |
| Observation area | Approx. 280 m$^2$ | Approx. 200 m$^2$ | Approx. 120 m$^2$ | Approx. 550 m$^2$ |
| Spatial features | Open space<br>Sufficient sunlight<br>High accessibility<br>Mainly stalls | Short corridor path<br>Low ceiling<br>Dark space<br>Mainly stalls | Short corridor path<br>Sufficient sunlight<br>Stalls and street shops | Semi-open shed space<br>High accessibility<br>Multiple types of activities<br>Mainly stalls |
| Photos of sites |  |  |  |  |

*3.2. Methodology*

The study was primarily divided into four phases. Phase 1 was a pilot study and field observation to select four sites, identify the spatial characteristics of each site, test hypotheses for the microclimate, and develop methods. Phase 2 was about the formal data collection; a mixed-methods approach was applied for the formal survey, including subjective questionnaires (n = 101), face-to-face interviews (n = 81), spatiotemporal behaviour mapping (n = 8455), and on-site observations. Three types of data were collected including data on microclimate, the visitors' behaviours in conjunction with their demographic background (such as activity types, genders, and age groups), and testimonies of people's on-site experiences. Phase 3 involved a data analysis (detailed in Section 4.3). This research was conducted in a public space and all personal information was anonymous; therefore, an ethics review was not necessary.

3.2.1. Subjective Questionnaires

A subjective questionnaire was used to obtain microclimate data. Common subjective microclimate survey indicators included "Feeling", "Temperature", "Sunshine", "Dryness", and "Wind" [7]. Based on the pilot study, "Visibility", "Noise", and "Smell" were identified as potential factors influencing visitor behaviour. Eight factors, comprising "Feeling", "Temperature", "Sunshine", "Dryness", "Wind", "Visibility", "Noise", and "Smell", were selected to represent microclimate. A total of 110 questionnaires were distributed across the four sites at different times and 101 valid questionnaires were returned.

The questionnaires were rated on a Likert scale: "Feeling" and "Smell" were set from "bad" to "good" on a five-point scale from 1 to 5, respectively; "Temperature", "Sunshine", "Dryness", and "Wind" were set from "low" to "high" on a five-point scale from 1 to 5, respectively; "Visibility" was likewise set from "Blur" to "Clear"; and "Noise" was set from "quite" to "loud" on a five-point scale from 1 to 5, respectively. The original questionnaire is attached to this article in the Supplementary Materials.

3.2.2. Face-to-Face Interviews

Face-to-face interviews were used to obtain information on where visitors came from, how often they came to the Antique Market, primary purpose of their visit, and their on-site experiences of the microclimate. The three main questions were as follows: Firstly, "why did you come here?"—i.e., their primary purposes and intended areas to visit. "How often do you come here?"—this question also included "Are you a local?" and, if they were not, asked where they were from. Finally, "how long do you stay in (or walk around) the market?" To increase efficiency, the maximum interview duration was maintained at 15 min.

The eighty-two sampled participants were interviewed on 13 and 18 July 2021, with a valid interview sample of eighty-one included in the analysis. The sampled participants were people passing through the four sites in the Panjiayuan Antique Market. The number of participants at the four sites correlated to their size, characteristics, and actual number of people attracted to the sites; for example, Site 4 was the largest and had the highest number of visitors, and therefore the highest number of participants were selected near Site 4 (The number of participants: n = 35). The other three sites were about the same size as each other, and there was not much difference in visitor numbers, and thus the number of participants selected near the other three sites was almost the same. The number of participants in Site 1 was 15, in Site 2 was 15, and in Site 3 was 16. The gender ratio of the participants was also taken into account. The pilot study and field observation showed that there were more male visitors in the Panjiayuan Antique Market, so there were more male than female participants selected (Male: 53, Female: 28).

Based on the frequency of visits, it was possible to determine whether the buyers were ordinary or regular visitors. The frequency of visits can be summarised as follows: (1) every day; (2) 1–2/week; (3) 1–2/month; (4) once every 1–2 months; (5) 2–3/year; (6) 1–2/year; (7) once every few years; and (8) for the first time, where (1)−(4) are defined as regular visits, (5)−(7) as occasional visits, and (8) as ordinary visitors. Classification of the frequency of visits revealed where visitors came from each of the four sites. Based on purpose of the visit, major activities of visitors were grouped into six categories: (1) targeted searching for treasures; (2) untargeted searching for treasures; (3) wandering (without the expectation of consumption); (4) meeting friends; (5) ghost market shopping; and (6) learning about antiques. Depending on the duration of the visit, these six activities could be divided into two major categories as walking and staying. The walking activities included (2) untargeted searching for treasures, (3) wandering, and (5) ghost market shopping. Staying activities included: (1) targeted searching for treasures, (4) meeting friends, and (6) learning about antiques. The classification of walking and staying activities was prepared for the following statistics of the number of people walking and staying per minute. The results of the interviews provided a general idea of the daily scenario in the Panjiayuan Antique Market in Beijing.

### 3.2.3. Spatiotemporal Behaviour Mapping

We used the spatiotemporal behaviour mapping (STBM) method developed by Sun et al. (2020) [33]. Briefly, at the four sites selected in this study, 13 photos were taken per hour (i.e., a photo was taken every 5 min at each site). Seven (every 10 min) were input to the GIS database, and remaining six were retained as backup. Observing from an elevated viewing position enabled an observation area of up to 1000–2000 m$^2$ to be covered. The recording periods included four different hours of the day as 10.00–11.00, 12.00–13.00, 16.00–17.00, and 19.00–20.00, which were considered representative of the daily rhythms of Panjiayuan Antique Market based on the pilot study. The STBM survey was conducted on 13 and 18 July 2021, on weekdays and weekends. This method can be used to roughly calculate the number of male and female visitors at each site and determine the relationship between the microclimate and the gender of visitors.

In addition, data on the number of walks per minute (NW) and number of stays per minute (NS) were obtained from on-site observations. NW were grouped into four levels: 0–3/min, 4–10/min, 11–20/min, and 21+/min. This indicated the percentage of people in the walking condition within the field of view per minute for a selected period. NS was categorised into four levels: 0/min, 1–3/min, 4–7/min, and 8/min. This indicated the percentage of people staying within the field of view per minute for a selected period.

### 3.3. Data Analysis

The general idea of the data analysis was to correlate the three databases obtained from the face-to-face interviews, subjective questionnaires, and STBM. Each of the three databases revealed different contents and meanings. The face-to-face interviews revealed

the individual profiles of visitors and their main activities in the market. The STBM database objectively revealed the characteristics and behavioural patterns of visitors in the market as a whole, to study and obtain the subjective perception of visitors regarding the environmental microclimate in the market. An analytical framework was established based on the three databases. The four main analyses were the impact of different microclimatic factors on visitors' subjective experience, the relationship between microclimate and people activities (these two analyses required correlating the face-to-face interview database with the subjective questionnaire database), and the relationship between microclimate and gender of visitors (this analysis needs the correlation of the subjective questionnaire database with the STBM database). The primary analytical methods used were analysis of means, analysis of variance (ANOVA), and linear regression analysis. The details of the analysis are as follows:

NVivo 11 was used to analyse the face-to-face interview material. First, interview transcripts were collected to summarise the general impressions of visitors and their behaviour. Second, the transcripts were repeatedly coded to extract specific visitor images (e.g., regular visitors, occasional visitors, ordinary visitors) and behavioural patterns (e.g., walking behaviour, staying behaviour). These findings were prepared for further analysis using other databases.

Data from 101 valid subjective questionnaires were entered into the SPSS Statistics 26.0 software. Mean value analysis was used to reveal the overall level of subjective perceptions of the microclimate in the Panjiayuan Antique Market. Using ANOVA (with 'openness of site' as X-classification and 'microclimatic factors' as Y-quantification) and linear regression analysis (with microclimatic factors and openness of site as independent variables, and subjective feeling as the dependent variable) were used to examine which microclimatic factors directly affected subjective feelings and which microclimatic factors differed significantly due to different levels of openness at the site.

The data from the STBM were entered into a GIS. These databases were used to determine the ratio of men to women at different times (week, weekend, and four time periods of the day) and locations (in the four selected sites).

ANOVA (with openness of site as X-classification, and the number of people walking and staying as Y-quantification) and linear regression analysis (with microclimatic factors and openness of site as independent variables, and number of those walking as the dependent variable) were used to reveal the types of activities that are affected by microclimate, and identify the microclimatic factors that had an effect. A comparative analysis was used to explain the relationship between changes in microclimate and sex ratios across time and sites.

## 4. Results

### 4.1. Impact of Microclimate on Visitors' Experience

Based on the result of ANOM (Table 2), the scores of "Feeling" of Site 2, Site 3, and Site 4 were higher than the median (the median value was 3, meaning moderate), while Site 1 was below the median (the mean value was 2.54). Site 2 had the best feeling, with a mean value of 3.29. Site 4 had the second-best feeling with a mean value of 3.2, and Site 3 had the third-best feeling with a mean value of 3.16. We found that "Visibility" and "Noise" primarily showed a wide range of trends, whereas other factors showed a consistent trend. Moreover, the mean value of "Visibility" was significantly higher than that of the other six factors, indicating that high "Visibility" was a more prominent feature of the sites, followed by "Noise", which also had a higher mean value, indicating that "Noise" likewise had a stronger impact on subjective feelings. In contrast, the mean value of "Wind" was significantly lower than that of the other factors, indicating that "Wind" has the weakest impact on the perception of microclimate, followed by "Sunshine", which also had a lower mean value, indicating that "Sunshine" also had a weaker influence at the sites.

**Table 2.** ANOM of microclimate factors.

|  | Feeling | Temperature | Sunshine | Dryness | Wind | Visibility | Noise | Smell |
|---|---|---|---|---|---|---|---|---|
| **Site 1** | 2.54 | 3.21 | 2.67 | 2.96 | 1.92 | 4.13 | 2.67 | 2.75 |
| **Site 2** | 3.29 | 2.96 | 2.5 | 2.63 | 2.33 | 3.83 | 3.25 | 3.38 |
| **Site 3** | 3.16 | 3.04 | 2.12 | 2.6 | 2.04 | 4.52 | 2.4 | 3.08 |
| **Site 4** | 3.2 | 3.16 | 2.6 | 2.8 | 2.24 | 3.24 | 3.6 | 2.6 |

The mean values of "Visibility" across the four sites were Site 3 > Site 1 > Site 2 > Site 4, in descending order, with more open sites having higher mean values of "Visibility" than the less open sites, which was consistent with common sense. However, it was not the case that the higher the "Visibility" value, the better the "Feeling". For example, Site 3 and Site 1 (MV = 4.52 and 4.13, respectively), which had higher mean values of "Visibility", had lower mean values of "Feeling" (MV = 3.16 and 2.54, respectively), while Site 2 and Site 4 (MV = 3.83 and 3.24, respectively), which had lower mean values of "Visibility", had higher mean values of "Feeling" (MV = 3.29 and 3.2, respectively). It can be inferred that the closer the mean value of "Visibility" is to the median level (note that it is not lower than the MV), the higher the mean value of "Feeling", suggesting that the feeling is better. This is inconsistent with the common sense. Therefore, we propose that in the market, it was not higher "Visibility" that was better, but rather the fact that people can obtain a better experience with a certain amount of visual interference or blockage within their field of vision.

"Noise" functioned similarly to "Visibility". The mean value of "Noise" was higher in less open sites than in more open sites, which was consistent with common sense. However, the lower the "Noise" level, the better the subjective feeling. In our study, the lower the mean value of "Noise", the lower the mean value of the "Feelings" was. It can also be inferred that the closer the mean value of noise is to the median level (note that it is not higher than the MV), the higher the mean value of "Feeling", which was likewise inconsistent with common sense. Therefore, we propose that in the market, it is not less noise that is better, but rather the fact that people can hear noisy background sounds that give them a better experience.

According to the ANOVA results (Table 3), "Visibility" and "Noise" were related to the physical characteristics of the sites. The degree of openness was the most obviously different physical characteristic among the four sites. Site 1 was an open space, Site 2 was a semi-open space with a low ceiling, Site 3 was an unroofed alley-like passage, and Site 4 was a covered semi-open space. Openness of site showed a 0.05 level of significance (*F* = 2.942, *p* = 0.037) for "Feeling"; openness of site showed 0.01 level significance (*F* = 6.4, *p* = 0.001) for "Visibility"; openness of site showed 0.01 level significance (*F* = 6.97, *p* = 0.000) for "Noise". Therefore, the openness of site showed significant differences in "Feeling", "Visibility", and "Noise", but not in "Temperature", "Sunshine", "Dryness", "Wind", and "Smell".

**Table 3.** ANOVA of microclimate factors.

| Openness of Site (MV ± SD) | Feeling | Temperature | Sunshine | Dryness | Wind | Visibility | Noise | Smell |
|---|---|---|---|---|---|---|---|---|
| **Site 1** (n = 25) | 2.60 ± 0.82 | 316 ± 0.94 | 2.60 ± 115 | 2.92 ± 1.15 | 2.00 ± 0.87 | 4.16 ± 0.85 | 2.64 ± 0.99 | 2.80 ± 1.66 |
| **Site 2** (n = 24) | 3.33 ± 1.05 | 2.96 ± 1.16 | 2.42 ± 1.25 | 2.63 ± 110 | 2.42 ± 0.88 | 3.79 ± 1.32 | 3.33 ± 1.40 | 3.38 ± 1.10 |
| **Site 3** (n = 27) | 3.11 ± 0.93 | 3.07 ± 0.92 | 2.26 ± 1.20 | 2.70 ± 1.07 | 2.00 ± 0.73 | 4.48 ± 0.80 | 2.44 ± 0.89 | 3.04 ± 1.34 |
| **Site 4** (n = 25) | 3.20 ± 0.91 | 316 ± 1.03 | 2.60 ± 1.38 | 2.80 ± 1.04 | 2.24 ± 0.93 | 3.24 ± 1.23 | 3.60 ± 0.87 | 2.60 ± 1.26 |
| *F* | 2.942 | 0.217 | 0.452 | 0.338 | 1.411 | 6.4 | 6.97 | 1.478 |
| *p*-Value | 0.037 * | 0.884 | 0.717 | 0.798 | 0.244 | 0.001 ** | 0.000 ** | 0.225 |

* $p < 0.05$, ** $p < 0.01$, MV: Mean Value, SD: Standard Deviation.

In addition, based on the results of the linear regression analysis (Table 4), an $R^2$ value of 0.217 indicated that the openness of site, "Temperature", "Sunshine", "Dryness", "Wind", "Visibility", "Noise", and "Smell" can explain 21.7% of the variation in feelings. The model passed the *F*-test ($F = 2.810$, $p = 0.006 < 0.05$), indicating that the openness of site and the seven microclimatic factors, at least one of which would have an influential relationship with feelings. The regression coefficient value for the openness of site was 0.172 ($t = 2.083$, $p = 0.040 < 0.05$), implying that openness had a significant positive effect on feelings. The regression coefficient for "Wind" was 0.337 ($t = 3.104$, $p = 0.003 < 0.01$), implying that "Wind" also has a significant positive effect on feeling, whereas other factors do not have an impact on feeling. However, there was almost no wind during the data collection period, which could explain the mean wind value of approximately two in the subjective questionnaire, which is below the median level. Therefore, in this study, "Wind" was not used as a factor affecting the subjective experience.

**Table 4.** Linear regression analysis of microclimate factors.

|  | **B** | ***t*-Value** | ***p*-Value** | **95%CI** |
|---|---|---|---|---|
| **Constant** | 1.516 | 2.043 | 0.044 * | 0.062~2.971 |
| **Openness of site** | 0.172 | 2.083 | 0.040 * | 0.010~0.334 |
| **Temperature** | 0.007 | 0.051 | 0.96 | −0.282~0.297 |
| **Sunshine** | −0.188 | −1.614 | 0.11 | −0.417~0.040 |
| **Dryness** | 0.111 | 1.108 | 0.271 | −0.085~0.306 |
| **Wind** | 0.337 | 3.104 | 0.003 ** | 0.124~0.550 |
| **Visibility** | 0.066 | 0.745 | 0.458 | −0.108~0.241 |
| **Noise** | −0.011 | −0.132 | 0.896 | −0.177~0.155 |
| **Smell** | 0.059 | 0.828 | 0.41 | −0.081~0.198 |
| **$R^2$** | | | 0.217 | |
| **Adjustment $R^2$** | | | 0.14 | |
| **$F$** | | | $F (9, 91) = 2.810$. $p = 0.006$ | |

Dependent variable: Feeling; CI: confidence interval, B: regression coefficient. * $p < 0.05$, ** $p < 0.01$.

### 4.2. Types of Visitors and Activities

Based on the database of face-to-face interviews (n = 81), ordinary tourists were the main component of the visitor mix in the Antique Market (Table 5). There were 19 regular, 23 occasional, and 39 ordinary visitors. The largest number of visitors (n = 35) had Site 4 as their main destination, whereas the other three sites had similar numbers of visitors (Sites 1, 2, and 3 had 15, 15, and 16 visitors, respectively). Site 4 was most attractive to ordinary tourists (n = 21), whereas Site 1 was most attractive to regular visitors (n = 9). It was assumed that the distribution of visitors is related to product type. The bracelets and beads traded primarily at Site 4 are inexpensive and widely available. This was consistent with ordinary tourists' goals. In Site 1, on the other hand, buyers use goods and antiques that are more expensive and require a higher level of background knowledge. This finding was consistent with the fact that most buyers there were regular visitors.

Walking was the main type of visitor activity (Table 5). Site 4 had the most targeted searching for treasure activity (n = 13), followed by Site 2 (n = 10), with 13 people looking for bracelets/beads and 10 people looking for used books. Site 1 had the highest number of untargeted searching for treasure visitors (n = 6), followed by Site 4 (n = 3) and Site 3 (n = 1). Site 4 had the highest number of wandering visitors (n = 16), followed by Site 3 (n = 10). Site 1 and Site 3 had a smaller proportion of visitors meeting friends, with 1 and 2 people respectively. Other activities included ghost market shopping and learning about antiques, which accounted for a smaller number of visitors; 5 and 2, respectively. Thirty-four people were classed as staying, while forty-seven people walked. The number of walking activities was consistently higher than that of staying activities. The only exception was site 2, where the number of people staying was higher.

**Table 5.** Types of visitors and activities in the four sites.

| The Frequency of Visits | Number of Visitors (n = 81) | | | |
|---|---|---|---|---|
| | Site 1 (n = 15) | Site 2 (n = 15) | Site 3 (n = 16) | Site 4 (n = 35) |
| **Regular visits (n = 19)** | **9** | **2** | **3** | **5** |
| Every day | 4 | 0 | 0 | 0 |
| 1–2/week | 1 | 1 | 2 | 3 |
| 1–2/month | 1 | 1 | 1 | 1 |
| Once every 1–2 months | 3 | 0 | 0 | 1 |
| **Occasional visits (n = 23)** | **3** | **4** | **7** | **9** |
| 2–3/year | 2 | 0 | 2 | 1 |
| 1–2/year | 1 | 3 | 4 | 2 |
| Once every few years | 0 | 1 | 1 | 6 |
| **Ordinary visitors (n = 39)** | **3** | **9** | **6** | **21** |
| For the first time | 3 | 9 | 6 | 21 |
| **Types of activities** | | | | |
| **Walking Activities (n = 47)** | **10** | **5** | **11** | **22** |
| Untargeted searching for treasure [1] | 6 | 1 | 1 | 3 |
| Wandering | 3 | 3 | 10 | 16 |
| Ghost market shopping [2] | 1 | 1 | 0 | 3 |
| **Staying Activities (n = 34)** | **6** | **10** | **5** | **13** |
| Targeted searching for treasure [3] | 4 | 10 | 2 | 13 |
| Meeting friends | 1 | 0 | 2 | 0 |
| Learning about antique | 1 | 0 | 1 | 0 |

[1] Untargeted searching for treasures is a unique expression of the Panjiayuan Antique Market. During the late Qing Dynasty, most of the antiques sold in the Panjiayuan Antique Market were stolen by servants of wealthy families or fallen aristocrats, and most of the antiques were genuine. Therefore, folk collectors had the expression "searching for treasure in Panjiayuan". Untargeted searching for a treasury implies enjoying the process of searching for a treasury more than buying a specific item. [2] Ghost market is also a unique term in the Panjiayuan Antique Market. As most goods in the early days were obtained by improper means, the host market traded at night and dispersed in the early morning, with people carrying lanterns around the ghost market. Currently, ghost markets are meant to be commercial activities that attract popularity. [3] Targeted searching for treasures is also a unique feature of the Panjiayuan Antique Market. This means that people with antique enthusiasm visit regularly and have regular preferences for items.

Walking activities were further divided into targeted and nontargeted walking. Except for Site 2, the number of people walking without a target at the other three sites was greater than those walking with a target. At Site 3, the two types of walking behaviour showed a significant difference. The number of people who walked non-targeted was four times that of people who walked, whereas at Site 1, the difference between the two types of walking behaviour was the smallest. The number of people undertaking non-targeted walking was twice that of people who walked.

In summary, the most important behavioural pattern among visitors was "tourist wandering". Most people were visiting a place for the first time and were in a state of novelty and curiosity regarding the environment. It is possible that the factors that influence

the subjective feelings of these visitors were features of the environment. Therefore, this study proposes the hypothesis that differences in the microclimates of the sites may affect the types of visitors and activities.

### 4.3. Microclimate and Activities

Based on the database of onsite observations (Table 6) at Sites 1 and 4, the number of walks was greater than the number of stays. The NW (The Number of Walking) at Site 1 was mainly 4–10/min and 11–20/min (37 and 29%, respectively), whereas the NS (The Number of Staying) was mainly 4–7/min (50%). NW was the most at 11–20/min (44%) at Site 4, while NS was the highest at 1–3/min and 4–7/min (both 40%). At Sites 2 and 3, the number of stays was not significantly different from the number of walks. The NW at site 2 was mainly 0–3/min and 4–10/min (46% and 33%, respectively), whereas the NS was mostly 4–7/min (38%). At Site 3, the NW was mostly 4–10/min (44%). The NS was mostly 8+/min and 4–7/min (36% and 24%, respectively). It should also be noted that NW = 4–10/min and NS = 4–7/min were the two most significant levels. While 4–10/min was the most prominent distribution level of walking, the level of 11–20/min were also more than 30% at sites 1 and 4. It can be concluded that more people were active at sites 1 and 4 than at sites 2 and 3.

**Table 6.** Distribution of people walking and staying.

| NW [1] | Site 1 | Site 2 | Site 3 | Site 4 |
|---|---|---|---|---|
| 0–3 | 13% | 46% | 36% | 8% |
| 4–10 | 37% | 33% | 44% | 24% |
| 11–20 | 29% | 21% | 16% | 44% |
| 21+ | 21% | 0 | 4% | 24% |
| **NS [2]** | | | | |
| none | 10% | 12% | 24% | 12% |
| 0–3 | 32% | 25% | 16% | 40% |
| 4–7 | 50% | 38% | 24% | 40% |
| 8+ | 8% | 25% | 36% | 8% |

**NW [1]**: number of walks (per min). This indicates the percentage of people in the walking condition within the field of view per minute for a selected period. **NS [2]**: number of stays (per minute). This indicates the percentage of people in the stay condition within the field of view per minute for a selected period.

An ANOVA was used, with the openness of site as X (classification) and NW and NS as Y (quantification), to analyse whether X and Y showed significance (Table 7). The openness of the site had a significant effect on NW at the 0.01 level ($F = 6.767$, $p = 0.001$), while it had no impact on NS. The means for NW were 2.60 and 2.96 for Site 1 and Site 4, respectively, and 1.92 and 1.93 for sites 2 and 3, respectively. It can be concluded that open and semi-open sites attracted more walking activity, while low-ceiling spaces and alley-like spaces attracted less.

**Table 7.** Analysis of variance of Walking and Staying activities.

| | Openness of Site (Mean Value ± Standard Deviation) | | | | |
|---|---|---|---|---|---|
| | Site 1 (n = 25) | Site 2 (n = 24) | Site 3 (n = 27) | Site 4 (n = 25) | *p*-Value |
| NW [1] | 2.60 ± 1.12 | 1.92 ± 0.83 | 1.93 ± 0.92 | 2.96 ± 1.10 | 0.001 ** |
| NS [2] | 2.60 ± 0.71 | 2.71 ± 1.00 | 2.89 ± 1.40 | 2.44 ± 0.82 | 0.454 |

** $p < 0.01$. **NW [1]**: number of walks (per min). This indicates the percentage of people in the walking condition within the field of view per minute for a selected period. **NS [2]**: number of stays (per minute). This indicates the percentage of people in the stay condition within the field of view per minute for a selected period.

In addition, based on the results of the linear regression analysis (Table 8), Visibility, Temperature, Sunshine, Dryness, Wind, Noise, and Smell were used as independent

variables, with NW as the dependent variable for the linear regression analysis. An $R^2$ value of 0.598 implied that Visibility, Temperature, Sunshine, Dryness, Wind, Noise, and Smell could explain 59.8% of the variation in NW. An *F*-test of the model revealed that it passed the *F*-test ($F$ = 3.617, $p$ = 0.014 < 0.05), which meant that at least one of the microclimatic factors had an effect on NW. This analysis revealed that visibility and noise had significant positive effects on the number of walks (per min). However, temperature, sunshine, dryness, wind, and smell did not positively affect the number of walks (per min).

**Table 8.** Linear regression analysis of microclimate factors and walking activity.

|  | B | *t*-Value | *p*-Value | 95%CI |
|---|---|---|---|---|
| Constant | −1.349 | −1.136 | 0.272 | −3.676~0.979 |
| Visibility | 0.538 | 2.544 | 0.021 * | 0.123~0.952 |
| Temperature | −0.554 | −2.089 | 0.052 | −1.075~−0.034 |
| Sunshine | 0.351 | 1.626 | 0.122 | −0.072~0.774 |
| Dryness | 0.062 | 0.414 | 0.684 | −0.230~0.353 |
| Wind | −0.033 | −0.136 | 0.894 | −0.502~0.437 |
| Noise | 0.685 | 4.131 | 0.001 ** | 0.360~1.011 |
| Smell | 0.013 | 0.102 | 0.92 | −0.240~0.266 |
| $R^2$ | | | 0.598 | |
| Adjustment $R^2$ | | | 0.433 | |
| *F* | | | $F$ (7, 17) = 3.617. $p$ = 0.014 | |

Dependent variable: Feeling; Cl: confidence interval, B: regression coefficient. * $p < 0.05$, ** $p < 0.01$.

### 4.4. Microclimate and Visitors' Gender

The study also found that female visitors are more sensitive to the microclimate of the environment and the characteristics of the site and have a higher need to feel safe than men. Based on the results of the STBM survey, the number of male buyers was greater than that of female buyers on both weekdays and weekends (Table 9). Site 2 had the greatest difference between male and female buyers, with the ratio of male buyers to female buyers on weekends at 3.61 and on weekdays at 3.01. Site 4 had the smallest difference between male and female buyers, with the ratio of male buyers to female buyers at 2.18 and weekends at 2.62. This suggested that site 2 was more attractive to male buyers, whereas the difference in attractiveness between male and female buyers at site 4 was not significant.

**Table 9.** Gender ratio of visitors and mean value of microclimate factors.

|  | Ratio of Gender | | | | | | Factors of Microclimate (Mean Value) | |
|---|---|---|---|---|---|---|---|---|
|  | Weekdays and Weekends (M/F) | | Four Time Points in a Day (M/F) | | | | | |
|  | Weekdays | Weekends | 10.–11. | 12.–13. | 16.–17. | 19.–20. | Visibility | Noise |
| Site 1 | 2.05 | 3.34 | 2.18 | 1.87 | 1.73 | 1.54 | 4.13 | 2.67 |
| Site 2 | 3.01 | 3.61 | 3.41 | 3.58 | 3.06 | 1.49 | 3.83 | 3.25 |
| Site 3 | 2.37 | 3.19 | 3.90 | 3.33 | 2.89 | 2.58 | 4.52 | 2.40 |
| Site 4 | 2.18 | 2.62 | 3.52 | 2.72 | 2.85 | 1.43 | 3.24 | 3.6 |

**M/F:** Male-to-female ratio.

Based on the ANOM, "Visibility" and "Noise" were the microclimatic factors that varied the most across the four sites. The higher the male-to-female ratio, the closer the mean values of "Visibility" and "Noise" were to the median. In contrast, the lower the male-to-female ratio, the greater the mean values of "Visibility" and "Noise" were from the median value.

At Site 2, the male-to-female ratio on weekdays and weekends was higher relative to other three sites (3.01, 3.61) and at four times of day (3.41, 3.58, 3.06, 1.49), and the mean values for "Visibility" and "Noise" were closer to the median level (3.83, 3.25). At Site 1, the ratio of men to women was lower on weekdays and weekends (2.05, 3.34) and at four times of the day (2.18, 1.87, 1.73, and 1.54), and the mean values for "Visibility" and "Noise"

were far from the median level (4.13, 2.67). At Site 3, the ratio of men to women was lower on weekdays and weekends (2.37, 3.19) and at four times of the day (3.90, 3.33, 2.89, and 2.58), and the mean values for "Visibility" and "Noise" were far from the median level (4.52, 2.40).

These data can be explained by the fact that women were more sensitive to microclimates compared to men. A higher M/F ratio indicated fewer women and more men. "Visibility" and "Noise" at this time were close to the median, indicating that people were close to being satisfied with the microclimate. The lower the M/F, the closer the number of men and women; that is, the number of women was relatively high, and both "Visibility" and "Noise" were far from the median, indicating that people were more sensitive to the microclimate. No further findings on the other effects of the microclimate on the gender of visitors are available at this time. However, the openness of these sites also affects female visitors' activities. For example, sites 1 and 2 were open and semi-open sites with a relatively high number of female visitors, whereas site 3 was an alley-like site with a narrower, more crowded space, and a relatively low number of female visitors.

## 5. Discussion

Thermal comfort in a microclimate is often considered to have the most significant impact on people's feelings and experiences [10,15]. However, in cultural consumption spaces, such as the Panjiayuan Antique Market, thermal comfort did not appear to have a decisive impact on people's perceptions, but other factors (such as "Visibility" and "Noise") did. The evaluation of these influences also showed some results that were not in line with common sense, such as "Visibility", where it might be supposed that the clearer the vision, the better the subjective feeling. Rather, people expected a certain amount of visual interference, obscuration, or confusion within the field of view and obtain a better sense of experience. A similar result was observed for the impact of "Noise" on subjective perception, initially presuming that the quieter the image, the better the experience. Instead, people can hear noisy background sounds and infer a more vibrant marketplace. This result is consistent with the face-to-face interview results, such as "the atmosphere is very good, the vendors are very warm and interesting, and there are many tourists" (N5), "unexpected, good feeling" (N33), "quite characteristic of the antique market" (N38). This finding suggested that specific places and scenarios should be considered when studying the impact of microclimates on people's experiences and behaviours. While outdoor thermal comfort is important, other factors also have a significant impact and correlation with people 's experiences and behaviours. Nevertheless, the "Visibility" and "Noise" proposed in this study did not suggest that there was still a correlation with people's experiences and behaviour in other cultural consumption spaces, but it suggests that in other types of outdoor spaces, all the impacts of microclimatic factors need to be considered.

The data from this study showed that the microclimate does not have a direct impact on walking activity, which is inconsistent with previous studies [7]. However, according to the results of the face-to-face interviews, some respondents reported that "it was too hot, so I could only walk around" (N60), "it was a bit noisy for children to walk around and I felt tired easily" (N17), "the smell was not good, I don't want to walk around anymore" (N21), and so on. These responses suggested that the microclimate influenced people's willingness to walk, and indirectly affected their walking behaviour. This finding was not reflected in the data analysis. In addition, stall owners of open and semi-open sites make spontaneous adaptations and amendments to their sites. For example, Site 1 is fully open, and stall owners cover this site with a huge, soft shed that keeps light and rain out. In the semi-open space of Site 4, similar strategies emerged from the stall sellers; for example, in the stalls on the side of the aisle, sellers would put up umbrellas to shade their exposed stalls and buyers. The spontaneous behaviour of these sellers improves the environment, allowing most visitors to ignore the discomfort caused by the microclimate. Therefore, a subjective questionnaire alone can yield data that ignore objective environmental conditions. Field measurements and software simulations are

required to support the subjective questionnaire [20] and explore the relationship between the real outdoor environment and the artificially changed microclimate environment.

In addition, the data showed that the microclimate did not have a direct impact on the distribution of men and women or the different ratios of visitors by gender. However, the results of the face-to-face interviews revealed that one of the reasons for the low ratio of walking activity by female visitors was that most of the female visitors were in an accompanying capacity; for example, "I came with my brother, I didn't do much shopping, I just sat and rested" (N7), "I accompanied my husband, I walked around a bit and then sat on a chair with my son" (N42), "I had a coffee with a friend, I wasn't shopping" (N52). Female visitors themselves did not show much interest in antique market wares, resulting in less walking activity by female visitors than by male visitors. Moreover, the majority of male visitors did not actively mention microclimate perceptions in the site when interviewed, while it was mostly women who actively mentioned microclimate perceptions, such as "the environment is noisy" (N16), "the market is a bit confusing, and it is easy to lose your sense of direction" (N27), and "The market feels a bit chaotic and hot" (N79). This finding confirms that female visitors are more sensitive to microclimates than men are, which influences their experience and behaviour towards the market.

A limitation of this study was the lack of application of instrumental measurements for microclimate. The reason for this was that in this study, we paid more attention to exploring the impact of people's subjective perceived microclimate on their experience and behaviour, as well as the impact of site characteristics and cultural context on subjective perceptions. However, if field measurements and software simulations are included in future studies, individual differences in perception of microclimate can be better identified. In addition, the duration of the present study was not long enough, it should be conducted over longer durations to obtain a relatively well-developed database in the future study.

## 6. Conclusions

Returning to the research questions for this study, our findings showed that "Visibility" and "Noise" directly influenced people's experiences and behaviour in the Panjiayuan Antique market. The major subjective perceptions were that the lower the "Visibility", and the higher the "Noise", the better the reported perception of the market from people. This conclusion was supported by the data from the Face-to-face interview question "How do you feel about the place?" Previous studies on the relationship between outdoor microclimates and human perceptions have primarily focused on thermal comfort or wind. Our findings confirmed that other microclimatic factors can also influence subjective perceptions. However, the microclimatic factors that influence subjective perceptions vary depending on the site characteristics. The impacts of "Visibility" and "Noise" in this study may not be evident in other markets. However, our study has revealed that people's subjective perceptions of microclimate were not reflected by objective conditions of microclimate, but were also influenced by specific situation, cultural and social contexts. Therefore, the microclimatic factors that have a dominant influence on subjective perception vary in different sites. In addition, results of the data analysis showed that women were more sensitive to changes in microclimate than men. Face-to-face interviews confirmed that female visitors were mostly accompanied by male visitors. Therefore, the attention of female visitors was easily drawn to the environment. This finding suggested that there is a difference in how genders perceive the microclimate, but that this is related to the social nature of the environment.

The findings of this study can be used to conduct a post-occupancy evaluation (POE) of an outdoor built environment, as impact of the microclimate on people's experiences and behaviour is ultimately attributed to the characteristics of the built environment. Our findings help to identify current problems in the built environment that have the potential to lead to poor perception and behaviour, and the comprehensive assessment of POE to suggest strategies for improving the built environment, so as to better guide design and avoid the huge waste of environmental and social resources that can result from faulty design. The unique contribution of this study was that vendors in the Panjiayuan Antique

Market spontaneously amended their nearby environments, potentially influencing the microclimate quality and enhancing the perception of visitors' experiences. This is a spontaneous POE (as opposed to systematic POE), which not only confirmed the importance of the microclimate in the POE of outdoor spaces, but also suggested that the spontaneous POE conducted by users has vital place in the theory of POE, which has not received sufficient attention. Additionally, the mixed method used in this study provided a broader and more detailed approach to POE data collection and analysis.

**Supplementary Materials:** The following supporting information can be downloaded at: https://www.mdpi.com/article/10.3390/buildings13051158/s1, The original questionnaire.

**Author Contributions:** M.H. and Z.S. conceived and designed the study; M.H., B.H. and Z.S. collected the data and prepared the original manuscript; M.H. and S.L. performed data analysis and edited the manuscript; Z.S. critically edited the manuscript. All authors have read and agreed to the published version of the manuscript.

**Funding:** This research was supported by Academic Initiation Programme for Young Scholars at Beijing Institute of Technology, grant number XSQD-202018003.

**Informed Consent Statement:** Informed consent was obtained from all subjects involved in the study.

**Data Availability Statement:** The data is unavailable due to privacy and ethical restriction.

**Acknowledgments:** We would like to express our gratitude to the volunteers from Beijing Institute of Technology who supported the data collection. We would like to thank the managers of Panjiayuan Antique Market for facilitating this research. This research was supported by Academic Initiation Programme for Young Scholars at Beijing Institute of Technology.

**Conflicts of Interest:** The authors declare no conflict of interest.

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
