# Peer review of "Impact of Microclimate on People’s Experiences and Behaviours in the Cultural Consumption Space: A Case Study of Panjiayuan Antique Market in Beijing, China"

_buildings, doi:10.3390/buildings13051158_

Round 1

Reviewer 1 Report

The authors investigated microclimate factors, subjective experiences and spatio-temporal behavior patterns in an antique market. This paper needs great improvement before it can be published. Some comments are listed here for reference.

1.      Few studies in the last three years are referenced. The references can’t reflect the latest   research findings in this field. Authors should pay more attention to the latest papers on microclimate and POE.

2.      Literature review needs improvement. The logic needs to be strengthened.

3.      There are 84 volunteers in this study. How were the visitors selected? Was the selection random or purposeful? And the demographic information of the volunteers is not given in the text.

4.      In section “Subjective Questionnaires”, it is suggested to list the specific questions in a table, so that the readers can have a more comprehensive understanding of the questionnaire and the experiment.

5.      In section “Face-to-face Interviews”, authors should explain the procedure and details of the experiment more clearly.

6.      The methods of data analysis are too homogeneous. More in-depth and multifaceted data analysis is needed. In addition, this paper lacks figures to show the results.

7.      Can the conclusions be applied to other antique markets or other places? If not, the findings of this paper lack universality and applicability.

8.      The overall language needs improvement. Authors need to correct the spelling and grammar mistakes carefully.

Author Response

Dear Reviewer, thank you very much for the comments you gave on this article, they have been very insightful and helpful. My specific reponses to each of the comments are as follows:

Point 1: Few studies in the last three years are referenced. The references can’t reflect the latest research findings in this field. Authors should pay more attention to the latest papers on microclimate and POE.

Response 1: I have added 19 new papers to the reference list, nine of these are studies from the last three years.

Most studies on microclimates have focused on the relationships between microclimates, urban morphology, and landscape greenery,…The outdoor thermal environment directly affects outdoor activities and use of outdoor space… However, people often have more varied thermal experiences and expectations. …Recent studies indicated that environmental psychology is an alternative way of understanding microclimate impacts. Namely, the microclimate factors might conceal different meanings and have various impacts on human behaviors in different situations and cultural backgrounds, which have not yet been sufficiently explored. …studies on the relationship between microclimates and human behaviour have focused on the impact of specific microclimatic factors on human behaviour, and the influence of spatial features of buildings on microclimate. …However, fewer studies have taken the time into accout.

Most of the recent studies on POE has focused on the application of multiple data forms or databases to optimise existing data collection and analysis approaches to achieve a more scientific assessment of the built environment. However, there is a lack of study on the collection of data on people moving around in small-scale outdoor spaces and the effects of time on human behaviour and microclimate.

Point 2:  Literature review needs improvement. The logic needs to be strengthened.

Response 2: The literature review has been restructured as a whole and revised as follows

The microclimate is an important part of the urban ecology, …, and is an vital factor in environmental equality[8]. (This text begins on line 46 of the revised version)

Microclimate characteristics can be reflected by physical parameters, …and the correlation between microclimate and outdoor thermal comfort [11]. (This text begins on line 58 of the revised version)

Most studies on microclimates have focused on the relationship between microclimates, …trigger sudden changes in pedestrian walking speeds within a space at the level of individual intentions [31]. (This text begins on line 68 of the revised version)

Point 3:  There are 81 volunteers in this study. How were the visitors selected? Was the selection random or purposeful? And the demographic information of the volunteers is not given in the text.

Response 3: Eighty-two sampled participants interviewed on 13th and 18th July 2021, with a valid interview sample of eighty-one. …so there were more male than female participants selected (Male: 53, Female: 28). (This text begins on line 172 of the revised version)

Point 4: In section “Subjective Questionnaires”, it is suggested to list the specific questions in a table, so that the readers can have a more comprehensive understanding of the questionnaire and the experiment.

Response 4: The questions for the “Subjective Questionnaires” are added in the Supplementary Materials. (This text begins on line 172 of the revised version)

Point 5: In section “Face-to-face Interviews”, authors should explain the procedure and details of the experiment more clearly.

Response 5: Face-to-face interview were used to obtain information on where visitors came from, …the maximum interview duration was maintained at 15 min.(This text begins on line 165 of the revised version)

Point 6: The methods of data analysis are too homogeneous. More in-depth and multifaceted data analysis is needed. In addition, this paper lacks figures to show the results.

Response 6: 3.3 Data Analysis was revised as follow:

The general idea of the data analysis was to correlate the three databases obtained from face-to-face interviews, Subjective Questionnaires and STBM. …The details of the analysis are as follows. (This text begins on line 223 of the revised version)

NVivo 11 was used to analyse the Face-to-face interview material. …These findings were prepared for further analysis using other databases. (This text begins on line 238 of the revised version)

Data from the 101 valid subjective questionnaires were entered into the SPSS software. …due to different levels of openness of the site. (This text begins on line 244 of the revised version)

The data from STBM were entered into a GIS. …and different space (in the four selected sites). (This text begins on line 252 of the revised version)

ANOVA (openness of site as X-classification and the number of walking and staying as Y- quantification) and… to explain the relationship between changes in microclimate and sex ratios across time and sites. (This text begins on line 255 of the revised version)

Point 7: Can the conclusions be applied to other antique markets or other places? If not, the findings of this paper lack universality and applicability.

Response 7: The overall revision of the first and second paragraph of section 6. Conclusion is as follows:

Returning to the research questions in this study, our findings showed that “Visibility” and “Noise” directly influenced people’s experiences and behaviour in the Panjiayuan Antique market. …However, the microclimatic factors that influence subjective perceptions vary depending on the site characteristics, specific situation, cultural and social contexts. …the microclimatic factors that have a dominant influence on subjective perception vary in different sites. (This text begins on line 520 of the revised version)

The findings of this study can be used to conduct a Post-occupancy Evaluation (POE) of an outdoor built environment, …Our findings help POE to suggest strategies for improving the built environment, to better guide design and avoid the huge waste of environmental and social resources that can result from faulty design. …This is a spontaneous POE… conducted by users has vital place in the theory of POE, … provided a broader and more detailed approach to POE data collection and analysis. (This text begins on line 540 of the revised version)

Point 8: The overall language needs improvement. Authors need to correct the spelling and grammar mistakes carefully.

Response 8: The overall language has been improved by professional editor.

Reviewer 2 Report

I suggest to authors to make a few improvements in the manuscript:

1) Methods - to add Questionnaires and content of Reviews in the supplementary material. It could be useful for other researchers to use i their research;

2) Methods - Did you have microclimate measurements in parallel with questionnaires? If yes, it would be good to put a new sub-chapter like Climate monitoring to explain in more details background of climate data (instruments, locations, parameters, quality of datasets, etc.). If not, than explain why monitoring on the field was not performed and is it a shortcoming of the research.

3) Conclusions - in 2-3 sentences to explain the applicability of this research, i.e. how can be useful for local community and useful in different climate change and environment strategies.

Author Response

Dear Reviewer, thank you very much for the comments you gave on this article, they have been very insightful and helpful. My specific reponses to each of the comments are as follows:

Point 1: Methods - to add Questionnaires and content of Reviews in the supplementary material. It could be useful for other researchers to use in their research;

Response 1: The questions for the “Subjective Questionnaires” are added in the Supplementary Materials.

Point 2: Methods - Did you have microclimate measurements in parallel with questionnaires? If yes, it would be good to put a new sub-chapter like Climate monitoring to explain in more details background of climate data (instruments, locations, parameters, quality of datasets, etc.). If not, then explain why monitoring on the field was not performed and is it a shortcoming of the research.

Response 2: A limitation of this study was the lack of application of instrumental measurements for microclimate. …we paid more attention to explore the impact of people's subjective perceived microclimate on their experience and behaviour, as well as the impact of site characteristics and cultural context on subjective perceptions. (This text begins on line 506 of the revised version)

Point 3: Conclusions - in 2-3 sentences to explain the applicability of this research, i.e. how can be useful for local community and useful in different climate change and environment strategies.

Response 3: The overall revision of the first and second paragraph of section 6. Conclusion is as follows:

Returning to the research questions in this study, our findings showed that “Visibility” and “Noise” directly influenced people’s experiences and behaviour in the Panjiayuan Antique market. …However, the microclimatic factors that influence subjective perceptions vary depending on the site characteristics, specific situation, cultural and social contexts. …the microclimatic factors that have a dominant influence on subjective perception vary in different sites. (This text begins on line 520 of the revised version)

The findings of this study can be used to conduct a Post-occupancy Evaluation (POE) of an outdoor built environment, …Our findings help POE to suggest strategies for improving the built environment, to better guide design and avoid the huge waste of environmental and social resources that can result from faulty design. …This is a spontaneous POE… conducted by users has vital place in the theory of POE, … provided a broader and more detailed approach to POE data collection and analysis. (This text begins on line 540 of the revised version)

Round 2

Reviewer 1 Report

Authors made major revisions to this paper, which greatly improved its quality. After the authors further check and correct the data and language, this paper can be published. For example, in Table 4, for Site 1, the numbers of “Untargeted searching for treasure”, “Wandering”, “Ghost market shopping” are 6, 3, 1, respectively. However, the sum of the three does not equal the number of “Walking activities”, which is 9. Authors need to carefully correct these details.

Author Response

Point 1: In Table 4, for Site 1, the numbers of “Untargeted searching for treasure”, “Wandering”, “Ghost market shopping” are 6, 3, 1, respectively. However, the sum of the three does not equal the number of “Walking activities”, which is 9. 

 Response 1: Dear Reviewer, thank you very much for your insightful comment. I have corrected the number of “Walking activities” for Site 1 and rechecked all the data in the tables carefully. The revised table is attached.

Reviewer 2 Report

I have no further comments.

Author Response

Comments and Suggestions from the Reviewer: I have no further comments.